# D-AR: Diffusion via Autoregressive Models

**Ziteng Gao**
Show Lab, National University of Singapore
gzt@outlook.com

**Mike Zheng Shou**[✉]
Show Lab, National University of Singapore
mike.zheng.shou@gmail.com

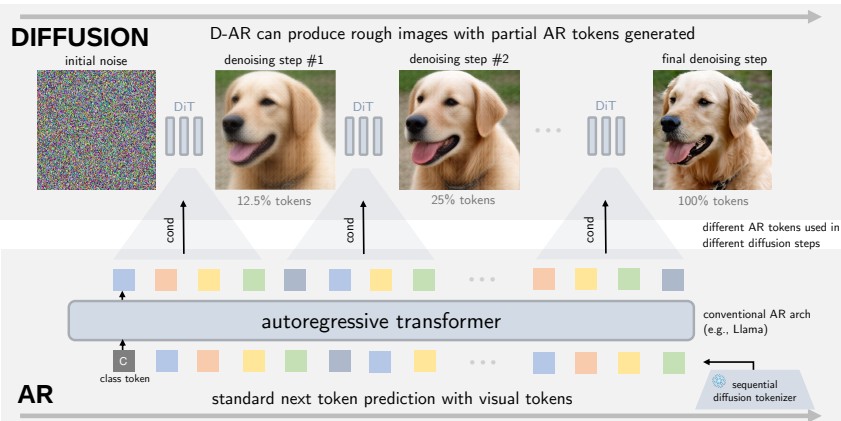

Figure 1: **Diffusion via autoregressive modeling (D-AR) framework** for visual generation. As the autoregressive transformer generates tokens, D-AR can simultaneously perform corresponding diffusion steps via token conditioning and jump-estimate target samples as rough previews effortlessly.

## ABSTRACT

This paper introduces Diffusion via Autoregressive (D-AR) models, a new paradigm recasting the pixel diffusion process as a vanilla autoregressive procedure in the standard next-token-prediction fashion. We start by designing the tokenizer that converts an image into the sequence of discrete tokens, where tokens in different positions can be decoded into different diffusion denoising steps in the pixel space. Thanks to the diffusion property, these tokens naturally follow a coarse-to-fine order, which directly lends itself to autoregressive modeling. Then, we apply standard next-token prediction to these tokens, without modifying any underlying designs (either causal masks or training/inference strategies), and such sequential autoregressive token generation directly mirrors the diffusion procedure in image space. That is, once the autoregressive model generates an increment of tokens, we can directly decode these tokens into the corresponding diffusion denoising step on pixels in a streaming manner. Our pipeline naturally reveals several intriguing properties, for example, it supports consistent previews when generating only a subset of tokens and enables zero-shot layout-controlled synthesis. On the standard ImageNet benchmark, our method achieves 2.09 and 2.00 FID using a 775M and 1.4B Llama backbone with 256 discrete tokens. We hope our work can inspire future research on unified autoregressive architectures of visual synthesis, especially with large language models. Code and models are available at https://github.com/showlab/D-AR.

## 1 INTRODUCTION

Autoregressive models, exemplified by large language models (LLMs) (Touvron et al., 2023; GPT-4-Team, 2024; Llama-3-Team, 2024), now underpin modern NLP, delivering state-of-the-art results

---

[✉]: Corresponding Author.

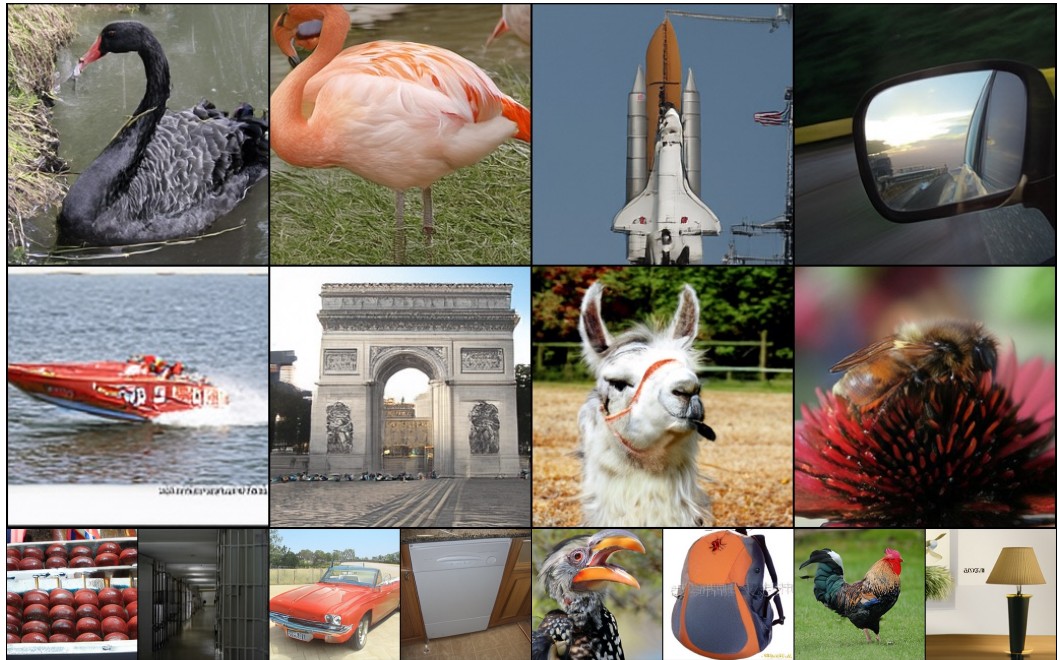

Figure 2: **Uncurated generated samples** from D-AR-XL with $256 \times 256$ resolutions (CFG=4.0).

with a simple next-token prediction objective. With widespread adoption of AR models, this simple next token prediction paradigm has established as the de facto standard in modern LLM systems and fostered software ecosystem for optimizing such training and inference pipelines (Shoeybi et al., 2020; Kwon et al., 2023; Zheng et al., 2024). The remarkable success of autoregressive models in language has also inspired exploration into visual generation tasks (Esser et al., 2021; Sun et al., 2024; Lee et al., 2022), with the broader goal of building unified frameworks of both vision and language (Team, 2024; Zhou et al., 2025; Xie et al., 2025; Ma et al., 2024b). However, unlike text, where sequential structure is naturally defined, images lack an inherently linear ordering, posing challenges for adapting such paradigm to vision modeling. Recent studies explore different visual orderings in autoregressive modeling (Tian et al., 2024; Pang et al., 2024; Yu et al., 2024a; Ren et al., 2025; Li et al., 2025). However, these approaches typically require significant modifications to the core mechanisms, often deviating from the standard next token prediction objective.

At the same time, modern vision generation pipelines are most dominated by diffusion paradigms (Song et al., 2021; Ho et al., 2020; Lipman et al., 2023), exemplified by several commercial systems (Ramesh et al., 2021; Labs, 2023; Podell et al., 2024). The diffusion pipelines excel at modeling continuous image signals: starting from random noise, they iteratively refine input through denoising to produce high-quality images. However, diffusion sampling requires many dense sequential dense denoising steps and such architectures pose challenges for seamless integration with LLMs and limit their potential in unified multi-modal systems.

In this paper, we aim to bridge the diffusion process and autoregressive modeling for visual generation, leveraging strengths from both paradigms. Importantly, we maintain a strict adherence to the standard next-token prediction paradigm and make no changes to the underlying autoregressive mechanism to "simulate" the diffusion process on images. To achieve this, we present the *sequential diffusion tokenizer* to reinterpret the diffusion process on raw image pixels as a sequence of coarse-to-fine discrete tokens. In this formulation, early tokens represent conditions in early diffusion steps from pure noise, whilst later tokens capture progressive steps over less noised inputs, leading to a naturally linearized decomposition of visual sequence. We design the diffusion model in the proposed tokenizer to be light and fast, i.e., with around 185M parameters and 8 diffusion steps without extra VAEs, and achieve 1.52 rFID on ImageNet (Deng et al., 2009) with a total budget of 256 discrete tokens. With this design, we can perform the diffusion process on image pixels via predicting next token in token sequence with the autoregressive mechanism unchanged. Therefore, we name

this framework as **D-AR** (**D**ffusion via **Auto**regressive) models. D-AR excels on the ImageNet class-conditioned generation benchmark. With the plain LLaMA backbone (Touvron et al., 2023) backbones, 775M and 1.4B D-AR models achieve the leading 2.09 and 2.00 gFID with a total of 256 tokens in the standard next-token-prediction AR regime. We hope our work can inspire future research on integrated multi-modal LLM architectures with native visual generation capabilities.

## 2 RELATED WORK

### 2.1 DIFFUSION AND AUTOREGRESSIVE MODELS

Diffusion models and autoregressive models are currently two main streams of modern generative modeling. Diffusion models (Ho et al., 2020; Song et al., 2021; Lipman et al., 2023; Liu et al., 2023), exemplified by several commercial text-to-image models (Labs, 2023; Ramesh et al., 2021), excel in generating high-quality visual content by iteratively denoising a sample from an initial noise. Though powerful in generating visually pleasing images, the diffusion process typically operates in a dense manner and requires significant sampling steps, which can be computationally expensive. Recent success in language modeling using autoregressive paradigm, especially large language models (Llama-3-Team, 2024; GPT-4-Team, 2024; Team, 2025; Bai et al., 2023), has inspired researchers to explore the potential of this paradigm in visual generation tasks due to its scalability and mature training and inference infrastructures. However, this adaptation raises several challenges, since images are not inherently discrete and linear structures like text. To this end, researchers use vector quantized autoencoders to quantize images into discrete latent codes (van den Oord et al., 2017; Esser et al., 2021) and use raster-scan order to model the image sequence (Sun et al., 2024; Team, 2024; Wang et al., 2024). Researchers have also found that image sequence ordering can be defined in various ways (Tian et al., 2024; Pang et al., 2024; Ren et al., 2025; Yu et al., 2024a), and the next-token prediction paradigm should be adapted to suit vision modeling accordingly.

Though sorts of visual autoregressive models have been proposed, the dominant role of diffusion models in visual generation tasks remains almost unchanged due to their outperforming capabilities at visual continuous signals. In this paper, we seek to bridge diffusion models and autoregressive models for visual generation and leverage the advantage of both sides, following previous efforts in this research line (Li et al., 2024; Gu et al., 2025a; Chen et al., 2024; Deng et al., 2024; Wu et al., 2024; Pan et al., 2025b; Ge et al., 2024; Zhou et al., 2025; Gu et al., 2025b). But different from these work, we strictly adhere to the standard next-token-prediction autoregressive paradigm with discrete inputs and outputs, and design diffusion in the tokenizer decoder in a sequential manner to tackle with visual continuous data.

### 2.2 VISUAL TOKENIZATION WITH DIFFUSION MODELS

How to encode images into sequences of discrete tokens and then effectively reconstruct pixels from them is a key design for visual generation in autoregressive models. Due to the vector quantization and downsampling operations, visual tokenization methods inevitably suffer from the loss of information and lead to suboptimal reconstruction quality, which researchers have put intensive efforts into improving (van den Oord et al., 2017; Esser et al., 2021; Yu et al., 2022; Lee et al., 2022). Concurrently, a research direction recently emerges on leveraging diffusion models to decode visual tokens back into image pixels (OpenAI, 2023; Zhao et al., 2024; Tang et al., 2024; Sargent et al., 2025; Tang et al., 2024). Specifically, these methods typically see discrete tokens as conditions in the diffusion process. By doing so, they offload visual ambiguity and fine details to the diffusion model and significantly improves the visual fidelity (Zhao et al., 2024; Chen et al., 2025; Sargent et al., 2025). Further work on this line argues that discrete tokens should focus on structural semantics of images and extract such semantics with flexible sequence length (Wen et al., 2025; Bachmann et al., 2025) by large latent diffusion models together with VAE (Kingma & Welling, 2014; Rombach et al., 2022).

To our best knowledge, our method is the first to propose the tokenizer to interpret the full diffusion process into the autoregressive sequential generation using the diffusion tokenizer. Our method is individually developed from related work, DDT-LLama (Pan et al., 2025a) and Selftok (Wang et al., 2025), which also uses a diffusion decoder to sequentialize tokens but in a reversed order or

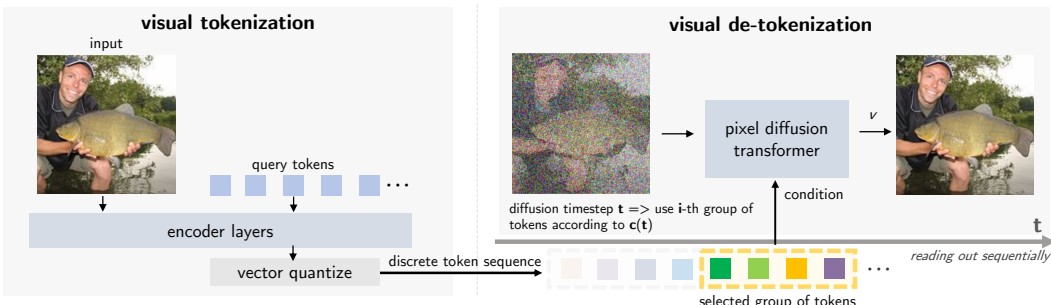

Figure 3: **Sequential diffusion tokenizer structure.** When training the tokenizer, the pixel diffusion transformer in the tokenizer decoder calculates the velocity loss with the selected group of tokens, $\mathbf{c}(t)$, as conditioning tokens.

a recursive way. Also, Pan et al. (2025a) and Wang et al. (2025) cannot represent diffusion steps as sequential AR generation process and therefore cannot decode with partial tokens generated by autoregressive models, marking a key distinction from our method and underlying motivation.

## 3 METHODS

A critical challenge in visual autoregressive modeling (Sun et al., 2024; Tian et al., 2024) is, for a long time, how to tokenize a 2D image into a sequence of discrete tokens since images are not inherently 1D linear structures like text. Though several works defined the ordering of image pixels (Pang et al., 2024; Tian et al., 2024; Yu et al., 2024a; Ren et al., 2025), they either introduce spatial inductive bias or require tailored autoregressive designs for vision, posing challenges on a unified autoregressive framework.

We propose a systematic solution to address this with Diffusion via Autoregressive models (D-AR), which recasts the image diffusion process as a fully autoregressive model in the standard next-token-prediction manner. The high-level idea is to *perform the diffusion process on pixels via autoregressive modeling*. To start, we design a *sequential diffusion tokenizer* that tokenizes images into sequences of 1D discrete tokens, which can be sequentially decoded as diffusion steps from the first to the end token. We apply standard next-token prediction on these tokens using a Llama decoder-only autoregressive backbones (Touvron et al., 2023), without modifying any AR architecture (either causal masks or training/inference designs) to generate images.

### 3.1 SEQUENTIAL DIFFUSION TOKENIZER

The sequential diffusion tokenizer is designed to tokenize images into 1D linearized discrete tokens in the ordering of progressive diffusion steps. The overall tokenizer structure is shown in Figure 3, akin to conventional visual tokenizers, which encodes images into latents, quantize them into discrete ones, and then decode them back into diffusion over pixels in an auto-encoding manner.

**1D encoding.** Similar to 1D tokenization approach (Yu et al., 2024b), the sequential diffusion tokenizer first encodes the image into a 1D sequence of discrete tokens using a transformer:

$$\mathbf{z} = [\mathbf{z}_1, \mathbf{z}_2, \ldots, \mathbf{z}_N] = \text{QUANT}(\mathcal{E}(\mathbf{I}, [\mathbf{q}_1, \mathbf{q}_2, \ldots, \mathbf{q}_N])), \tag{1}$$

where $\mathbf{I}$ is the input image, typically patchified as a set of patch tokens, $\mathcal{E}$ is the transformer encoder (Vaswani et al., 2017), $\text{QUANT}(\cdot)$ is the vector quantizer (van den Oord et al., 2017), and $[\mathbf{q}_i]$ are learnable query tokens, where $N$ is the total number of queries. In this step, we do not impose a specific ordering on the resulting 1D token sequence, which we will further focus on below.

**Sequential diffusion decoding.** We propose the sequential diffusion decoder to decode 1D quantized token sequence into consecutive diffusion steps on image pixels. The diffusion decoder is a diffusion transformer (Peebles & Xie, 2023), which takes tokens in different positions in the sequence as conditions in different diffusion steps. Here, flow matching loss with velocity prediction,

a simplified variant of diffusion families (Liu et al., 2023; Lipman et al., 2023; Ma et al., 2024a), is used to train the diffusion decoder. The loss is defined as:

$$\ell_{\text{fm}} = \mathbb{E}_{t,\mathbf{x}_0,\mathbf{x}_1} \left[ \|\mathbf{v}_t - \mathcal{D}_{\text{FM}}(\mathbf{x}_t, t, \mathbf{c}(t))\|_2^2 \right], \tag{2}$$

where the flow interpolant is defined as:

$$\mathbf{x}_t = t\mathbf{x}_1 + (1-t)\mathbf{x}_0, \quad \mathbf{v}_t = d\mathbf{x}_t/dt = \mathbf{x}_1 - \mathbf{x}_0, \tag{3}$$

$$\mathbf{x}_0 \sim \mathcal{N}(0,1), \quad \mathbf{x}_1 = \mathbf{I}, \quad t \in [0,1]. \tag{4}$$

With this notation, $\mathbf{x}_0$ at timestep $t = 0$ represents pure noise and $\mathbf{x}_1 = \mathbf{I}$ at $t = 1$ represents the real data sample. During inference, samples can be generated by solving ordinary differential equation (ODE) from $t = 0$ to $t = 1$ when the condition schedule $\mathbf{c}(t)$ is given.

The condition schedule $\mathbf{c}(t)$ is a set of quantized tokens $\mathbf{z}_i$ used as conditions in the diffusion decoder at timestep $t$. To enable the sequential decoding property, we design the condition schedule $\mathbf{c}(t)$ to start from the first token $\mathbf{z}_1$ and reach the last token $\mathbf{z}_N$ as the flow matching timestep $t$ progresses from 0 to 1. In preliminary experiments, we find that multiple $\mathbf{z}_i$ for a specified timestep is crucial for good performance. We thus first group consecutive tokens $\mathbf{z}_i$ into $K$ groups, $\{\mathbf{g}_1, \mathbf{g}_2, \ldots, \mathbf{g}_K\}$, each group $\mathbf{g}_i$ with $N/K$ tokens. The condition schedule is then defined as:

$$\mathbf{c}(t) = \mathbf{g}_{\lceil t' \cdot K \rceil}, \qquad t' = t/(t + (1/\beta) * (1-t)), \tag{5}$$

where $t'$ is the shifted timestep and $\beta$ is a control parameter. When $\beta = 1$, time ranges are evenly split regarding the condition group $\mathbf{g}_i$. The higher $\beta$ values lead to denser tokens as conditions over early diffusion steps, which we find empirically beneficial for reconstruction quality.

**Discussion.** One can view the 1D sequence of tokens as the "proxy" of the underlying diffusion procedure on pixels controlled by conditioning tokens $\mathbf{c}(t)$. With sequential diffusion decoding, we can decode increments of AR tokens into consecutive diffusion sampling steps on pixels in the streaming way when reading out tokens sequentially. This token order is naturally linearized by the diffusion process, where early tokens represent conditions needed in early diffusion steps ($t \to 0$) over noisy inputs, often low-frequency spatial layout. Later tokens describe the information needed in later steps ($t \to 1$) over less noisy inputs, typically localized details or structures (Rissanen et al., 2023). This coarse-to-fine token ordering is well-suited for autoregressive modeling, as shown in experimental section. Also, by the diffusion decoder, the tokenizer decoder can delegate ambiguous details to diffusion and thus focus on semantics (Hudson et al., 2024).

## 3.2 AUTOREGRESSIVE MODELING

Once we have the linearized sequence of discrete tokens by our proposed tokenizer, we can apply standard autoregressive next token prediction to model the image generation process:

$$p_\theta(\mathbf{z}) = \prod_{i=1}^{N} p_\theta(\mathbf{z}_i | \mathbf{z}_1, \ldots, \mathbf{z}_{i-1}), \tag{6}$$

where $\theta$ is the AR model parameters and one can use simple cross entropy loss to optimize parameters. In this paper, we resort to the decoder-only transformer architecture (Touvron et al., 2023; Sun et al., 2024) for autoregressive modeling.

**Vanilla vision autoregressive modeling.** General autoregressive modeling assumes a linear ordering of data elements, which is hard to define in images. By using tokens produced by the sequential diffusion tokenizer, D-AR keeps the same discrete inputs and outputs, attention masks/kernels, loss functions, and inference logistics as standard AR models

## 3.3 DIFFUSION VIA AUTOREGRESSIVE MODELS

The presented framework, diffusion via autoregressive models, simply consists of the sequential diffusion visual tokenizer and the Llama decoder-only transformer on discrete token sequences. Note here that the sequential diffusion tokenizer directly operates on raw pixels and do not require extra VAEs (Kingma & Welling, 2014; Rombach et al., 2022).

**Markovian diffusion procedure via vanilla autoregressive models.** As the name implies, sequential generation in the D-AR framework directly corresponds to diffusion procedure on image pixels via the bridge of token conditioning. When we are generating a sequence of tokens, we can perform the diffusion sampling on pixels simultaneously whenever we have condition tokens needed at diffusion timestep $t$ ready, i.e., $\mathbf{c}(t)$. Since the diffusion is only controlled by autoregressive models via condition tokens, we do not break the Markovian convention of diffusion models, different from a conceptually related work (Gu et al., 2025a). Therefore, D-AR can leverage advantages of both diffusion and autoregressive worlds:

1. **KV cache-friendly inference:** as the D-AR framework uses autoregressive decoder-only transformers on token sequences, it natively supports KV cache-friendly fast inference;

2. **Streaming pixel decoding and consistent previews at no extra costs.** We can perform diffusion steps on pixels instantly whenever we have needed tokens ready in a streaming manner. Also, since the diffusion decoder is directly operating on pixels, we can use the diffusion property to jump-estimate the target and generate consistent previews effortlessly;

3. **Zero-shot controlled synthesis.** As the token sequence is linearized by diffusion, we can simply condition several prefix tokens to control the visual generation without finetuning.

## 4 IMPLEMENTATIONS

**Sequential diffusion tokenizer architecture.** For the encoder in diffusion tokenizer, we mainly follow the design of 1D tokenizer (Yu et al., 2024b) to use the transformer encoder layers jointly processing image patches and learnable query tokens. We apply a causal mask to query tokens to enforce the basic causality on queries but allow both query tokens and image tokens to attend to arbitrary image tokens. As default, we set the number of queries $N = 256$, input patch size $p = 16$, the dimension of transformer $d = 768$, and the transformer layer $L = 8$. Following (Sun et al., 2024), we use the vanilla vector quantization with $\ell_2$-normalized codebook entries, configured with codebook size $n_e = 16384$ and dimension $d_e = 8$. We expect better performance with more advanced quantization approaches (Mentzer et al., 2024; Yu et al., 2023) but leave for future work.

We design the diffusion decoder as the diffusion transformer architecture (Peebles & Xie, 2023; Ma et al., 2024a) but on raw pixel patches, which integrates zero-initialized adaptive layer normalization (AdaLN)(Perez et al., 2018). To condition the diffusion decoder with condition tokens $\mathbf{c}(t)$, we use the cross attention layer on patch tokens to attend to condition tokens and take attention output as the input of the AdaLN, together added by the time $t$ embedding. The diffusion transformer decoder is configured moderately with $L_d = 12$ layers, $d_d = 768$ hidden dimension, and patch size $p_d = 8$, resulting in a total parameter of 185M.

We add causal decoder transformer layers on encoded tokens $\mathbf{z}$, after the vector quantization and before diffusion decoding, to produce $\mathbf{z}'$ for more nonlinearity. We configure it as the same as the transformer encoder. Note that these decoder transformer layers with causal masks do not break the causality of the token sequence. The total parameter of the sequential diffusion tokenizer is 300M.

**Training sequential diffusion tokenizer.** Training diffusion models on raw pixels with few inference steps is a challenging task (Hoogeboom et al., 2023; 2024), even with the strong image encoded conditions (Zhao et al., 2024). To enable few-step inference and speed up the convergence, we use the perceptual matching loss based on LPIPS (Zhang et al., 2018; Zhao et al., 2024) and representation alignment (REPA) loss (Yu et al., 2024c) together with flow matching (2) and vector quantization loss to train the sequential diffusion tokenizer:

$$\ell_{\text{tokenizer}} = \ell_{\text{fm}} + \ell_{\text{VQ}} + \lambda_1 \ell_{\text{LPIPS}} + \lambda_2 \ell_{\text{repa}}, \tag{7}$$

where we assign $\lambda_1 = 0.5$ and $\lambda_2 = 0.5$. We do not use adversarial matching loss (Zhao et al., 2024) in our training since we observe the instability and over-saturation issue.

In a training forward pass, we first encode an image into a quantized token sequence and use transformer decoder layers to compute $\mathbf{z}'$. Then we randomly sample a flow matching timestep $t \in [0, 1]$, determine which group $\mathbf{g}_i$ of $\mathbf{z}'$ should be used as conditions for diffusion decoder according to the condition schedule $\mathbf{c}(t)$, and compute the final loss $\ell_{\text{tokenizer}}$.

**Sampling with sequential diffusion tokenizer.** Given the token sequence, either encoded from images or generated from autoregressive modeling, we can perform the flow matching sampling by reading out tokens in the sequential order based on the condition schedule $\mathbf{c}(t)$. For simplicity and efficiency, we design the default sampling schedule to use each condition group exactly once, that is, to bind the number of sampling steps to the number of condition groups $K$ and use a timeshifted schedule in the reversed form of (5), following (Esser et al., 2024):

$$t_i = \frac{i/K}{(i/K) + \beta * (1 - i/K)}, \quad i = 0, 1, \ldots, K-1. \tag{8}$$

This sampling schedule results in denser early sampling steps when $\beta > 1$ and we default set $\beta = 2$ and $K = 8$ for sampling efficiency, resulting in each conditioning group with $N/K = 32$ tokens. Again, for efficiency, we do not use classifier-free guidance (CFG) (Ho & Salimans, 2022) in diffusion sampling decoding steps.

**AR models.** Our AR model architecture is exactly the same as Llama decoder-only transformers, which are with RMSNorm (Zhang & Sennrich, 2019) and SwiGLU (Shazeer, 2020). Note that since tokens by sequential diffusion tokenizer are inherently one-dimensional, we apply the original 1D RoPE (Su et al., 2024), rather than 2D RoPE, in attention layers as positional embedding. The class conditions, e.g., image labels, are injected as a single prefix token following (Sun et al., 2024). We do not use AdaLN in our AR models. Classifier-free guidance on logits is used during AR inference. We mainly experiment three variants of D-AR models, D-AR-{L, XL, XXL}, with 343M, 775M, and 1.4B parameters respectively, also following (Sun et al., 2024). To generate an image, D-AR models first produce a sequence of tokens conditioned on the given label in the standard token-by-token manner with KV cache enabled. In pace with sequential generation, we can decode tokens generated into diffusion sampling steps on pixels either concurrently or offline.

## 5 EXPERIMENTS

**Experimental Setup.** We conduct D-AR experiments on the ImageNet $256 \times 256$ class-conditional generation benchmark (Deng et al., 2009). The sequential diffusion tokenizer is trained on the ImageNet training set with a batch size of 1024, Adam optimizer (Kingma & Ba, 2015) of learning rate $2 * 10^{-4}$ and a total of 210K iterations till convergence, together with an exponential moving average with a 0.999 decay rate. The training procedure took around 5 days on 16 A100 GPUs to finish. We follow the training recipe (Pang et al., 2024) to train D-AR autoregressive models with a batch size of 1024 for 300 epochs. We use AdamW optimizer (Loshchilov & Hutter, 2017) with learning rate $4 * 10^{-4}$, $(\beta_1, \beta_2) = (0.9, 0.95)$ and weight decay of 0.05. The learning rate is decayed to $1 * 10^{-5}$ linearly within the last 50 epochs, following (Pang et al., 2024). The performance of D-AR is evaluated in terms of FID (Heusel et al., 2017), Inception Score (Salimans et al., 2016), precision and recall scores, following the standard ADM evaluation pipeline (Dhariwal & Nichol, 2021). For the reconstruction performance of the sequential diffusion tokenizer, we mainly investigate the reconstruction FID (rFID) on the ImageNet validation 50K set.

### 5.1 RESULTS

**Tokenizer results.** We investigate the key component of our D-AR framework, i.e., the sequential diffusion tokenizer. In Table 1, we compare our sequential diffusion tokenizer with the conventional LlamaGen tokenizer, which has the same budget of 256 tokens and the same vector quantization configuration, as strong baselines. Despite having more parameters (300M versus 72M), which is

| tokenizer | #tokens | codebook size | rFID↓ |
|---|---|---|---|
| RQ-VAE (Lee et al., 2022) | 256 | 16384 | 3.20 |
| Titok-S (Yu et al., 2024b) | 128 | 4096 | 1.71 |
| LlamaGen (Sun et al., 2024) | 256 | 4096 | 3.02 |
| LlamaGen (Sun et al., 2024) | 256 | 16384 | 2.19 |
| ours | 256 | 4096 | 1.84 |
| ours | 256 | 16384 | 1.58 |

Table 1: **Reconstruction results** on ImageNet validation 50K samples with 256 discrete tokens. We also finetune our sequential diffusion tokenizer with smaller codebook size, 4096, and compare with LlamaGen tokenizer counterpart.

| steps | 4 | 8 | 8, Adams 2nd | 12 | 16 |
|---|---|---|---|---|---|
| rFID↓ | 2.35 | 1.58 | 1.52 | 1.73 | 1.93 |

Table 2: **Different sampling configurations** on our sequential diffusion tokenizer. Adams 2nd refers to the two step Adams–Bashforth solver Bashforth & Adams (1883), while others use Euler.

mainly due to the pixel diffusion decoder, our sequential diffusion tokenizer achieves better reconstruction fidelity and is more endurable to smaller codebook size.

We also study different sampling configurations of the proposed sequential diffusion tokenizer in Table 2, where we vary the sampling steps and flow matching ODE solver. We use Adams–Bashforth solver for flow matching with 8 steps as it provides clearer samples without increasing numbers of function evaluations (NFEs) on the diffusion decoder.

**System-level comparison.** To compare with state-of-the-art methods, we experiment with D-AR models on the ImageNet $256 \times 256$ class-conditional generation benchmark. Following common practice (Li et al., 2024; Pang et al., 2024), the linear CFG schedule is used in D-AR (1.1→8.0 for D-AR-L, 1.1→10.0 for D-AR-XL, and 1.1→11.0 for D-AR-XXL). In Table 3, D-AR models achieve the leading level of performance in their parameter count regions. Among vanilla AR models in the strict next-token-prediction manner, D-AR-XL achieves 2.09 FID with 775M parameters, outperforming LlamaGen-XXL and even competing with IBQ-XXL 2.1B. D-AR-XXL achieves a state-of-the-art FID of 2.00 on ImageNet for vanilla AR models with 1.4B parameters.

Recent attempts to incorporate diffusion into autoregressive models, such as CausalFusion, DART-FM, and MAR, have also shown highly competitive results. However, they require significant modifications in the autoregressive framework to tackle continuous-valued inputs and outputs of images. In contrast, D-AR maintains the vanilla autoregressive mechanism with favored performance. DDT-Llama (Pan et al., 2025a) reported its 6.1 FID on ImageNet 256 but without mentioning parameter counts (2B or 8B), therefore we do not compare it in the table.

Table 3: System-level comparison on class-conditional generation over 50K samples on $256 \times 256$ ImageNet benchmark. Note that #params in the table only counts in AR model parameters and our tokenizer is with 300M parameters. MAR is difficult to categorize into mask-based or tailored AR methods.

| type | method | #params | FID↓ | IS↑ | Prec↑ | Rec↑ |
|---|---|---|---|---|---|---|
| diffusion | DiT-XL (Peebles & Xie, 2023) | 675M | 2.27 | 278.2 | 0.83 | 0.57 |
| | SiT-XL (Ma et al., 2024a) | 675M | 2.06 | 270.3 | 0.82 | 0.59 |
| mask-based | MaskGIT (Chang et al., 2022) | 227M | 6.18 | 182.1 | 0.80 | 0.51 |
| | TiTok-S-128 (Yu et al., 2024b) | 287M | 1.97 | 281.8 | - | - |
| | MAR-L (Li et al., 2024) | 479M | 1.78 | 296.0 | 0.81 | 0.60 |
| | MAR-H (Li et al., 2024) | 943M | 1.55 | 303.7 | 0.81 | 0.62 |
| tailored AR | VAR-d24 (Tian et al., 2024) | 1.0B | 2.09 | 312.9 | 0.82 | 0.59 |
| | VAR-d30 (Tian et al., 2024) | 2.0B | 1.92 | 323.1 | 0.82 | 0.59 |
| | RAR-L (Yu et al., 2024a) | 461M | 1.70 | 299.5 | 0.81 | 0.60 |
| | RAR-XL (Yu et al., 2024a) | 955M | 1.50 | 306.9 | 0.80 | 0.62 |
| | RandAR-L (Pang et al., 2024) | 343M | 2.55 | 288.82 | 0.81 | 0.58 |
| | RandAR-XL (Pang et al., 2024) | 775M | 2.22 | 314.21 | 0.80 | 0.60 |
| | RandAR-XXL (Pang et al., 2024) | 1.4B | 2.15 | 321.97 | 0.79 | 0.62 |
| | DART-FM (Gu et al., 2025a) | 820M | 3.82 | 263.8 | - | - |
| | CausalFusion-XL (Deng et al., 2024) | 676M | 1.77 | 282.3 | 0.82 | 0.61 |
| vanilla AR | LlamaGen-L (Sun et al., 2024) | 343M | 3.07 | 256.06 | 0.83 | 0.52 |
| | LlamaGen-XL (Sun et al., 2024) | 775M | 2.62 | 244.08 | 0.80 | 0.57 |
| | LlamaGen-XXL (Sun et al., 2024) | 1.4B | 2.34 | 253.90 | 0.80 | 0.59 |
| | IBQ-XL (Shi et al., 2024) | 1.1B | 2.14 | 278.99 | 0.83 | 0.56 |
| | IBQ-XXL (Shi et al., 2024) | 2.1B | 2.05 | 286.73 | 0.83 | 0.57 |
| | stronger LlamaGen-L (Pang et al., 2024) | 343M | 2.20 | 274.26 | 0.80 | 0.59 |
| | stronger LlamaGen-XL (Pang et al., 2024) | 775M | 2.16 | 282.71 | 0.80 | 0.61 |
| | **D-AR-L (ours)** | 343M | 2.44 | 262.97 | 0.78 | 0.61 |
| | **D-AR-XL (ours)** | 775M | 2.09 | 298.42 | 0.79 | 0.62 |
| | **D-AR-XXL (ours)** | 1.4B | 2.00 | 300.56 | 0.79 | 0.63 |

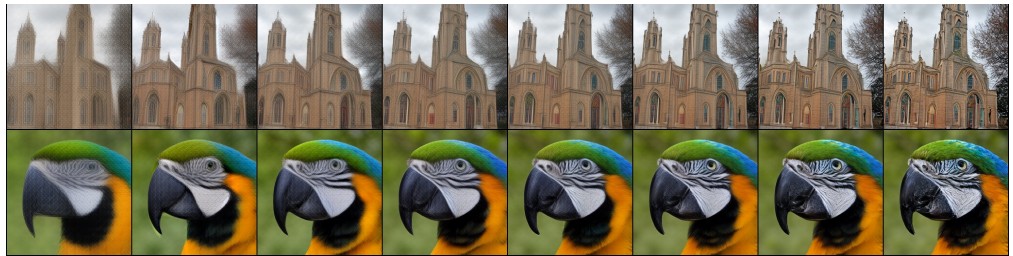

Figure 4: **Consistent previews as generation trajectories** for every increment of 32 tokens (a group). Note that these previews can be generated in a streaming manner with AR tokens partially generated.

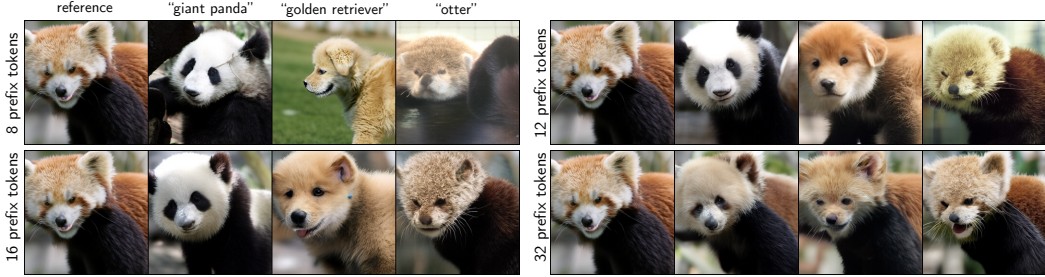

Figure 5: **Zero-shot layout-controlled synthesis** with different prefix tokens and varying labels.

**Coarse-to-fine diffusion order for AR modeling.** The order linearized by our proposed sequential diffusion decoder is naturally coarse-to-fine, which we argue is good for AR modeling. Here, we ablate this point by feeding the *reversed* token sequence into D-AR-L modeling by the sequential diffusion tokenizer, which is coresponding *fine-to-coarse* visual AR process, with the strict fair experiment setting. We name this *reversed* D-AR-XL and the generated sequence is then reversed again and decoded by the tokenizer decoder to get final image pixels. We searched multiple CFG schedules and the best result by *reversed* D-AR-XL is 4.17, which lags much behind the coarse-to-fine D-AR-L 2.44. More results can be found in the appendix. This comparison indicates that the diffusion induced coarse-to-fine order is the key to good visual autoregressive modeling, which is in line with Tian et al. (2024).

**Consistent previews and generation trajectories.** As discussed in Section 3.1, the sequential diffusion tokenizer can generate consistent previews of generated images when partial tokens are generated, inherited from the diffusion property to jump-estimate the target $\hat{\mathbf{x}}_1 = (1 - t)\mathbf{v}_t + \mathbf{x}_t$ for every sampling timestep $t$. As our diffusion model is on raw pixels, this operation takes almost no extra cost. We visualize these previews in Figure 4, which are consistent with final samples. These previews can also be interpreted as generation trajectories of our autoregressive model and inherently follow a coarse-to-fine progression (Rissanen et al., 2023).

**Zero-shot layout-controlled synthesis.** We also investigate the zero-shot layout-controlled synthesis with D-AR, where several prefix tokens are given and fixed, in Fig 5. Thanks to the linearized structure by the diffusion decoder, we can generate plausible images with reference layouts conditioned on reference prefix tokens and given labels, without specific finetuning. As more prefix tokens are provided, layout control becomes stronger, while label-relevant information increasingly concentrates on fine-grained details such as fur textures. We include more ablation studies and qualitative results in the appendix.

# 6   CONCLUSION

In this paper, we present Diffusion via Autoregressive models (D-AR), a framework to bridge pixel diffusion and autoregressive modeling for visual generation. With the linearized sequence of discrete tokens by the presented sequential diffusion tokenizer, we can perform vanilla autoregressive process in the standard next token prediction fashion. Thus, the AR sequence generation process in the D-AR framework directly mirrors consecutive diffusion denoising steps on pixels. Experiments on the standard ImageNet benchmark shows that D-AR can generate high-quality images as a vanilla autoregressive model, together with several properties from both autoregressive and diffusion worlds.

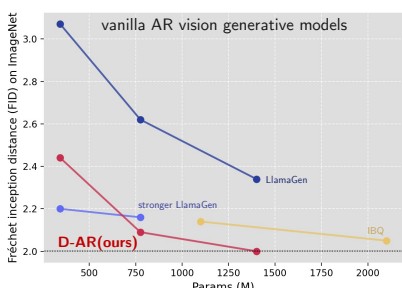

Figure 6: Vanilla AR comparison for ImageNet generation.

## ACKNOWLEDGMENTS

This project is supported by the National Research Foundation, Singapore under its NRFF Award NRF-NRFF13-2021-0008. We would like to thank Zhan Tong, Tong He, and Shuai Wang for helpful discussions and comments.

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

# A APPENDIX

## A.1 LLM USAGE

We use LLM to aid or polish writing, more specifically, to review and revise typos and grammar in this submission. We confirm that we fully reviewed the LLM-generated revisions and that the final revised text authentically reflects our original expression and ideas.

## A.2 DETAILED ARCHITECTURE OF SEQUENTIAL DIFFUSION TOKENIZERS

**Vector quantization.** We follow LLamaGen (Sun et al., 2024) to set up the vanilla vector quantization (van den Oord et al., 2017) as well as its loss: $\ell_{\text{VQ}} = ||\text{sg}[f] - z||_2^2 + \beta||f - \text{sg}[z]||_2^2$, where $\text{sg}[\cdot]$ is the stop gradient operator and $\beta = 0.25$. We do not impose the entropy loss on codebook learning.

**Transformer architecture.** In our sequential diffusion tokenizer, we adopt the transformer architecture with vanilla LayerNorm (Ba et al., 2016) and SiLU activation function (Elfwing et al., 2018). We also apply QK normalization (Henry et al., 2020) in attention computation for training stability. For tokens with explicit spatial locations, e.g., those patchified from images in the tokenizer encoder or in diffusion transformer, we apply the 2D RoPE (Su et al., 2024) in attention to encode spatial relations. For those who do not have 2D inherent locations, i.e., 1D query tokens in the transformer encoder and decoder, we simply disable rotation in RoPE by using the identity matrix.

## A.3 DETAILED EVALUATION OF D-AR MODELS

Table 4: D-AR with different CFG schedules. The value 1.0 indicates disabling CFG.

| model | CFG schedule | FID↓ | IS↑ | Prec↑ | Recall↑ |
|---|---|---|---|---|---|
| D-AR-L | 1.0 | 7.43 | 117.60 | 0.71 | 0.63 |
| | 1.5 | 3.50 | 245.22 | 0.83 | 0.54 |
| | 1.75 | 4.70 | 291.76 | 0.86 | 0.50 |
| | 1.1→8.0 | 2.44 | 262.97 | 0.78 | 0.61 |
| D-AR-XL | 1.0 | 5.11 | 145.78 | 0.73 | 0.64 |
| | 1.5 | 3.39 | 276.37 | 0.84 | 0.55 |
| | 1.1→10.0 | 2.09 | 298.42 | 0.78 | 0.62 |

**CFG schedules.** In the main paper, we present the performance of D-AR-L and D-AR-XL with linear CFG schedule, following RandAR (Pang et al., 2024). Note that previous work also explore customized CFG schedule for better performance, as a common practice on ImageNet (Li et al., 2024; Pang et al., 2024; Yu et al., 2024a). We report D-AR models results with different CFG strategies in Table 4. The models here are exactly the models in the main paper in Table 3. We do not use top-p, top-k, and temperature in our sampling in the main paper and appendix.

Table 5: D-AR-L jump-estimation results with partial AR tokens.

| #AR tokens | 64 | 128 | 192 | 256 |
|---|---|---|---|---|
| #diffusion steps | 2 | 4 | 6 | 8 |
| FID↓ | 7.38 | 3.94 | 2.93 | 2.44 |
| IS↑ | 165.25 | 227.74 | 257.08 | 262.97 |
| Prec↑ | 0.74 | 0.78 | 0.80 | 0.78 |
| Recall↑ | 0.48 | 0.54 | 0.57 | 0.61 |

**Partial AR tokens results.** In the main paper, we have visualized the diffusion target sample estimation with partial AR tokens generated. We here report quantitative results by D-AR-L in Table 5.

### A.4 TOKENIZER ABLATIONS

Due to the limited computation resource, we design a lightweight version of our proposed sequential diffusion tokenizer with 113M parameters (we change the dimension in the transformer to 512 and the depth of diffusion transformer to 8) and train for 50K iterations with 256 batch size. This ablation training typically takes about 8 hours to complete on 4 A100s.

Table 6: Effects of $\beta$ on tokenizer training.

| $\beta$ | 1 | 2 | 4 |
|---|---|---|---|
| rFID↓ | 39.75 | 28.65 | 27.10 |
| codebook utilization↑ | 97.8 | 99.4 | 99.8 |

**Ablations on $\beta$ in conditioning schedules.** The control parameter, $\beta$, in the conditioning schedule, also acts as timeshift parameter in the diffusion procedure in our sequential diffusion tokenizer. Since we are operating on pixels, we set $\beta$ to 2 as default. Here we investigate different $\beta$ on tokenizer training in Table 6. We can see that there is a large gap between $\beta = 1$ and $\beta = 2$ in reconstruction FID as well as in coodebook utilization, while $\beta = 2$ and $\beta = 4$ matches closer. These empirical results show that early diffusion need denser steps as well as AR tokens as conditioning on diffusion in the pixel space.

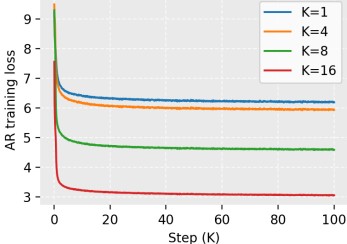

| $K$ | rFID↓ | gFID↓ |
|---|---|---|
| 1 | 8.37 | 34.85 |
| 4 | 9.66 | 34.54 |
| 8 | 14.18 | 35.69 |
| 16 | 22.58 | 41.20 |

Figure 7: Effects of the group $K$ on tokenizer and AR training. For fair comparison, for $K < 8$ variants, we use 8 diffusion sampling steps to decode images. rFID refers to sequential diffusion tokenizer reconstruction FID and gFID refers to D-AR-B generation FID at 100K iterations.

**The numbers of conditioning group $K$.** In this ablation, we stretch the sequential diffusion tokenizer training to longer 100K iterations. The number of conditioning group $K$ decides how many tokens are feed into pixel diffusion model per diffusion step, $N/K$. We investigate the effects of $K$ in the Table 7. The sequential diffusion tokenizer with single group $K = 1$ with multiple sampling steps degrades into conventional tokenizers with diffusion decoder (Sargent et al., 2025; Chen et al., 2025), which denoises an image with full token sequence on every timestep. This $K = 1$ setup does not yield a linearized ordering of visual tokens and lacks the sequential nature central to our approach.

For reconstruction FID here, it is reasonable and expected for small group number variants to perform better, since the number of conditioning tokens per denoising step become more as $K$ decreases, therefore reducing bottleneck. In the other side, $K = 16$ enforces the diffusion-induced linearized order most strongly, but came out with the worst reconstruction FID.

We also train a D-AR-B with 111M parameters for 100K iterations with a batch size of 1024 with tokens by these tokenizers. In the training loss curve in Figure 7, we can find that the large group number $K$ facilitates AR training, which we believe the stronger coarse-to-fine order is more well-suited for autoregressive modeling. Interestingly, although the reconstruction FID with $K = 8$ falls behind $K = 4$ and 1 variants, the generation FID achieved by D-AR-B models remains comparable. This suggests that more strongly linearized token sequences (higher $K$) can be better for autoregressive generation modeling even if they degrade reconstruction. For this reason, we adopt $K = 8$ as our default: it offers a moderate trade-off between reconstruction fidelity and a linearized structure

that benefits AR modeling, and as demonstrated in our main experiments, our $K = 8$ sequential diffusion tokenizer can eventually achieve competitive reconstruction performance.

Table 7: reversed D-AR-L with different CFG schedules. The value 1.0 indicates disabling CFG.

| model | CFG schedule | FID↓ | IS↑ | Prec↑ | Recall↑ |
|---|---|---|---|---|---|
| D-AR-L | 1.0 | 7.43 | 117.60 | 0.71 | 0.63 |
| | 1.5 | 3.50 | 245.22 | 0.83 | 0.54 |
| | 1.75 | 4.70 | 291.76 | 0.86 | 0.50 |
| | 1.1→8.0 | 2.44 | 262.97 | 0.78 | 0.61 |
| *reversed* D-AR-L | 1.0 | 11.22 | 96.23 | 0.68 | 0.62 |
| | 1.5 | 4.17 | 238.05 | 0.84 | 0.50 |
| | 1.75 | 5.83 | 292.94 | 0.88 | 0.44 |
| | 1.1→8.0 | 9.79 | 417.78 | 0.90 | 0.40 |
| | 8.0→1.1 | 21.15 | 320.15 | 0.85 | 0.15 |

## A.5 REVERSED D-AR

We here include more results by the *reversed* D-AR-L with different CFG configurations in Table 7. Note that the *reversed* D-AR-L training strictly follows the normal D-AR-L training setting, except for input token ordering. For the same CFG setting, the *reversed* D-AR-L falls behind the normal D-AR-L by a large margin.

## A.6 MORE VISUALIZATIONS

**Tokenizer reconstruction results.** We also present reconstruction samples from our sequential diffusion tokenizer (rFID = 1.52) in Fig 9. As observed, fine details are not strictly reconstructed, which is mainly attributed to the inherent stochastic and denoising nature of the diffusion process. Since our primary objective is to model image generation rather than achieve exact pixel-level reconstruction, this trade-off is acceptable and consistent with our diffusion tokenizer design.

**Generation trajectories.** We show more generation trajectories as well as previews in Fig 10. Our D-AR models follow coarse-to-fine generation with consistent previews with final targets.

**Zero-shot layout-controlled synthesis.** As discussed in the main paper, we can simply condition on prefix tokens to generate layout-following images in a zero-shot manner. We here show more zero-shot layout-controlled generated samples by fixing different numbers of prefix tokens and varying labels in Fig 11.

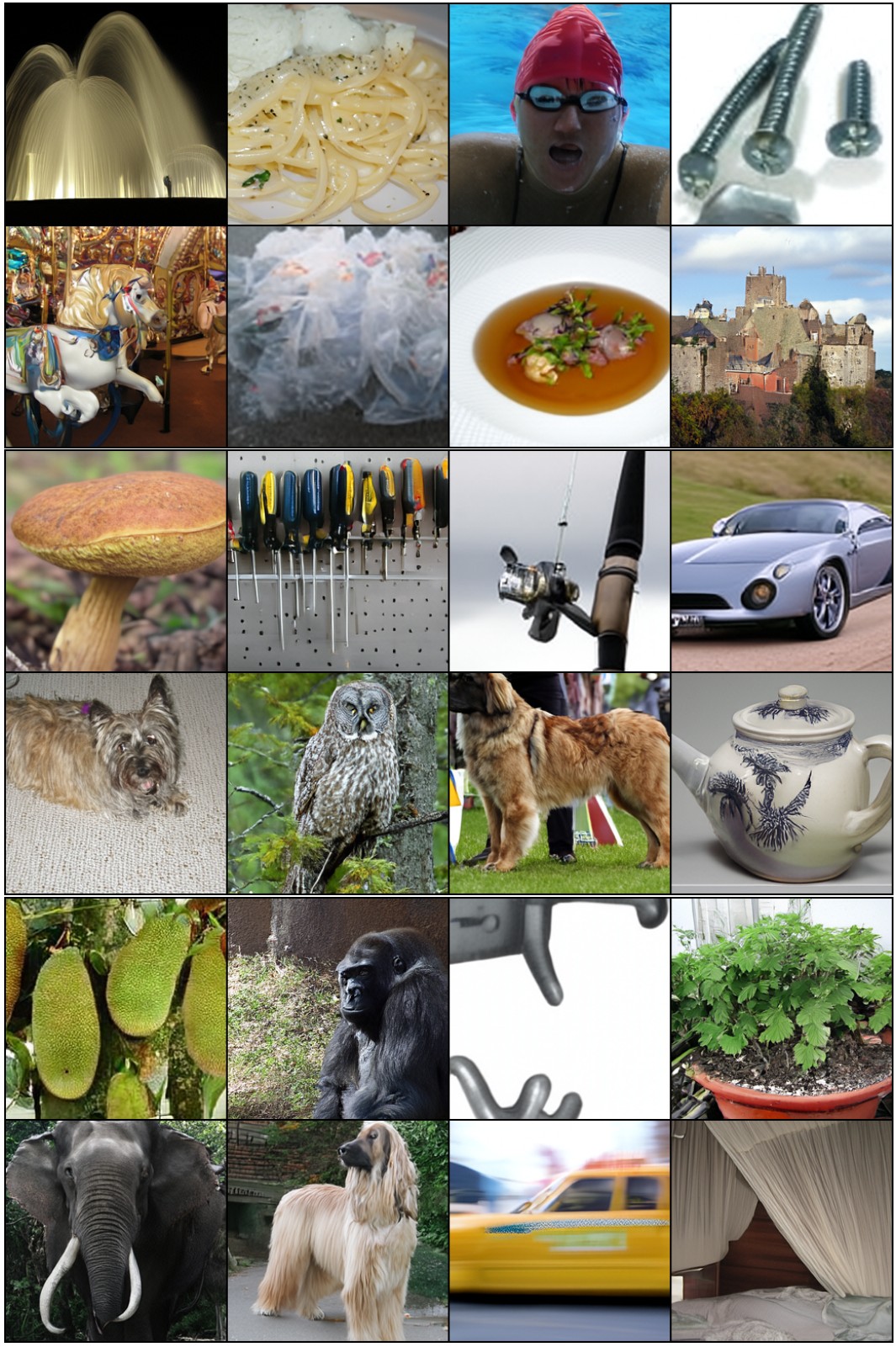

Figure 8: **Uncurated generated samples** by D-AR-XL with random labels and CFG=4.0.

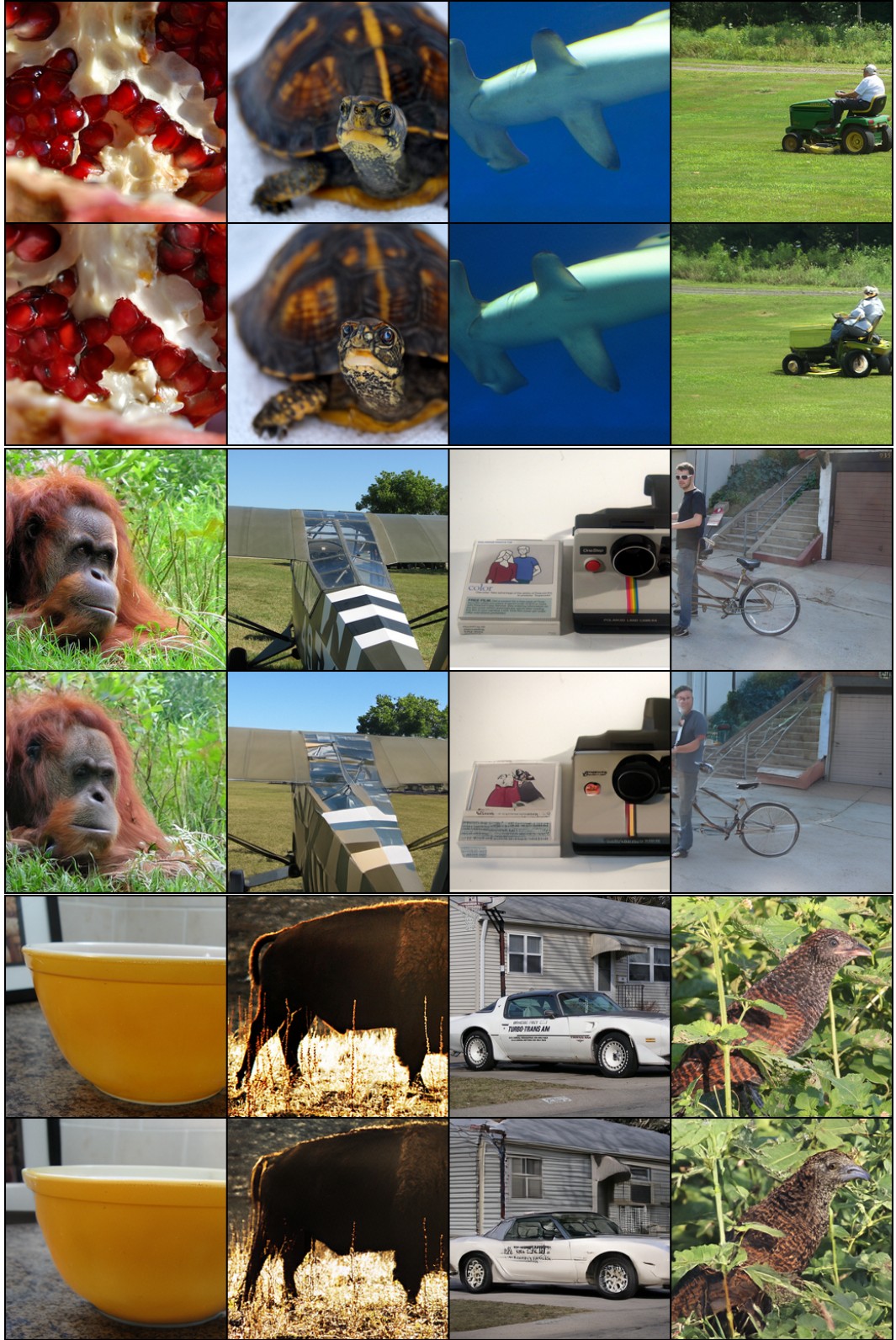

Figure 9: **Reconstruction results** with samples from the ImageNet validation set. Each pair of rows shows: first row — input; second row — reconstruction.

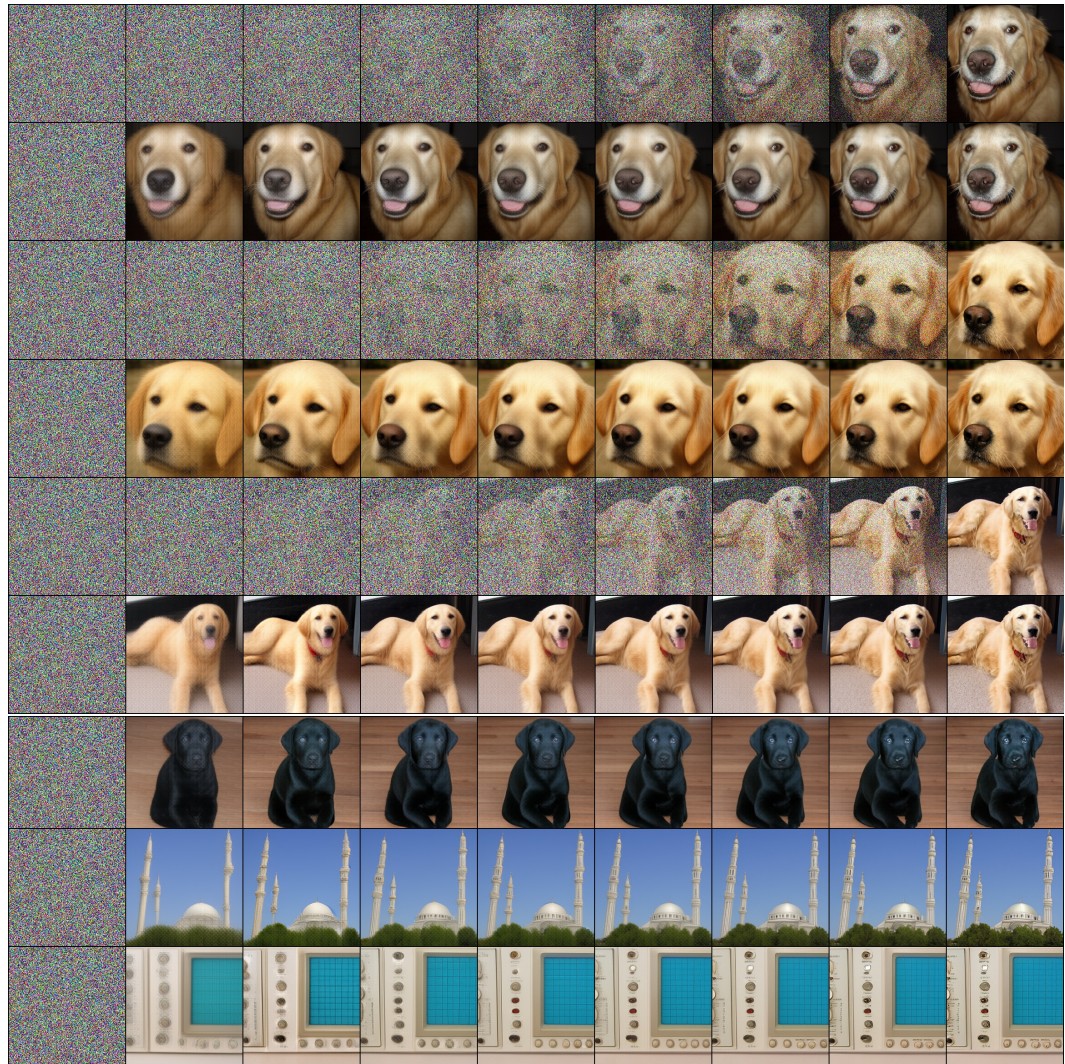

Figure 10: **Generation trajectory and previews** at each diffusion sampling step by D-AR-L.

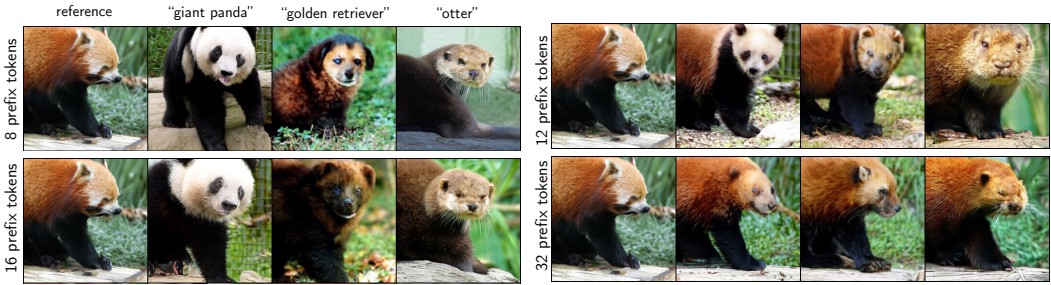

Figure 11: Zero-shot layout-controlled synthesis.

