# OpenReview forum: "D-AR: Diffusion via Autoregressive Models"
_ICLR.cc/2026/Conference — ICLR 2026 Poster_

### Official Review · Reviewer_iSt8 · 2025-10-25

**Soundness:** 3
**Presentation:** 3
**Contribution:** 3
**Rating:** 6
**Confidence:** 4

**Summary:**

This paper introduces "D-AR" (Diffusion via Autoregressive), a novel and elegant framework that reframes the image diffusion process as a standard autoregressive (AR) next-token prediction task.

The core of this work lies in a "Sequential Diffusion Tokenizer." Unlike traditional tokenizers that linearize an image spatially (e.g., raster-scan), this tokenizer encodes an image into a 1D sequence of **discrete tokens** (e.g., 256 tokens) that are ordered according to the **diffusion process itself** (coarse-to-fine).

The generation process is as follows:

1. A standard AR Transformer (e.g., Llama) uses standard next-token prediction to generate this discrete token sequence.

2. The tokenizer's decoder (a pixel-space Diffusion Transformer) uses these AR-generated tokens as the condition for its own denoising steps.

3. Critically, groups of tokens in the sequence correspond to specific diffusion steps. For example, the first 1-32 AR-generated tokens are used to condition step 1 of the diffusion; tokens 33-64 are added to condition step 2, and so on.

The authors claim SOTA results (2.00 FID) in the "vanilla AR models" category on the ImageNet 256x256 benchmark. The framework also naturally supports "streaming previews" (e.g., generating a 25% rough image after 25% of tokens are generated) and "zero-shot layout control" by fixing the initial (coarse-grained) prefix tokens.

**Strengths:**

**Conceptual Elegance and Novelty**: The main strength is conceptual. Reframing the entire multi-step diffusion process into a single standard AR sequence prediction problem is a highly elegant and novel "unification" of the two dominant generative paradigms (AR and Diffusion).

**Coarse-to-Fine Tokenization**: This is a key insight: linearizing an image based on "diffusion steps" (temporal, coarse-to-fine) rather than "space" (raster-scan). The ablation study in Section 5.1 (2.44 FID for forward order vs. 4.17 FID for reverse order) provides strong evidence for this design choice.

**SOTA Performance in "Vanilla AR" Category**: The paper achieves excellent results (2.00 FID with a 1.4B model, Table 3) within its defined category ("vanilla AR models"). It clearly and significantly outperforms the previous SOTA in this category, LlamaGen (2.34 FID).

**Zero-Shot Controllability**: The demonstration of zero-shot layout control by simply fixing prefix tokens (which correspond to early, coarse diffusion steps, Fig 5) is a powerful emergent property that validates the design's self-consistency.

**Weaknesses:**

**Potentially Misleading "SOTA" Classification**: The paper's headline result (2.00 FID) is only SOTA when compared to "vanilla AR models." As Table 3 shows, this performance still lags significantly behind "tailored AR models" (e.g., RAR-XL @ 1.50 FID), "mask-based models" (e.g., MAR-H @ 1.55 FID), and hybrid models (e.g., CausalFusion-XL @ 1.77 FID). Claims of "leading performance" should be more precise.

**Hidden Tokenizer Complexity**: The framing of the AR model as "vanilla" is debatable, as it outsources enormous complexity to the tokenizer, which is far from a simple VQ-GAN. The tokenizer is a 300M parameter model containing an encoder, a VQ, and a 185M parameter pixel-space Diffusion Transformer (DiT) decoder. This tokenizer is, in itself, a powerful generative model.

**Unaddressed Latency & Scalability Concerns**: The framework presents a dual-latency bottleneck (N-step AR generation + K-step Diffusion decoding) and was only tested on low-resolution ImageNet. Furthermore, the reliance on pixel-space operations (to enable previews) suggests a fundamental scalability challenge for high-resolution or T2I tasks, which is not discussed.

**Limited Evaluation Scope**: The entire paper is benchmarked only on ImageNet 256x256 class-conditional generation. This is a very constrained dataset that fails to test the model's ability to handle complex, open-ended text prompts (T2I) or scale to high resolutions (1024x1024), which are standard for SOTA diffusion models.

**Questions:**

To strengthen the paper, it is highly recommended that the authors address the following key questions with experiments:

**Question 1: Can a direct inference latency benchmark (e.g., sec/image) be provided?**

The paper claims "KV cache-friendly" inference but seems to have an inherent dual-latency challenge (N AR steps plus K Diffusion steps). How does the true wall-clock inference speed of D-AR (N=256, K=8) compare to (1) a standard 8-step DiT and (2) a standard VQ-GAN AR model (N=256)?

**Question 2: How is this coarse-to-fine tokenization paradigm expected to scale to high-resolution (1024x1024) T2I tasks?**

The model was only tested on ImageNet 256. Does a 256-token sequence have sufficient bandwidth to capture both the semantics of a complex text prompt and the fine details of a 1024x1024 image? Or would this require a much longer token sequence (e.g., N=1024), further exacerbating the AR latency bottleneck?

**Question 3: The ablation on K (number of groups) in Figure 7 seems counter-intuitive. Can this tradeoff be clarified?**

Figure 7 shows that K=16 (strongest linearization) gives the best AR training loss, yet the worst rFID and gFID. This seems to contradict the argument that a stronger coarse-to-fine order is better for AR modeling. Conversely, K=1 (no linearization) has poor AR loss but better rFID. Please clarify this critical tradeoff.

**Question 4: What is the distribution of "work" between the 1.4B AR model and the 300M Tokenizer?**

The 185M Diffusion decoder in the tokenizer is a powerful generator on its own. How much does the final image quality depend on the AR model's accuracy? For example, what is the FID if ground-truth tokens are provided for the first 50% of the sequence (z_1...z_128) and the AR model only generates the last 50% (z_129...z_256)?

Furthermore: If the AR model only generates the first 50% of tokens (z_1...z_128), but the DiT decoder uses only these tokens as the condition for the entire K=8 diffusion process (i.e., no new tokens are provided for t>0.5) while t progresses normally, can the model still produce an acceptable image? This would help clarify how much critical information is in the last 50% of tokens.

**Question 5: Is the "streaming preview" advantage overstated compared to standard diffusion?**

The paper claims its preview is a key advantage, but this is debatable. Standard diffusion models (pixel or latent) also produce coarse outlines from noise in early steps (e.g., step 4 of 20). The unique strength seems to be the link between "AR sequence progress" and "image clarity," rather than the preview capability itself.

**Question 6: Does fine-tuning this way waste the pre-trained LLM's capabilities?**

A selling point is the use of a standard Llama architecture. However, the model is fully fine-tuned in Stage 2 to predict visual tokens. To what extent does this process "destroy" the LLM's original, powerful language understanding and reasoning abilities? If the linguistic capabilities are lost, what is the true advantage of using a pre-trained Llama over a Transformer trained from scratch? Would a unified multimodal model (e.g., training on a mix of text and image tokens) be a superior path?

---

> ### Author Response · Authors · 2025-11-27
>
> We here would like to appreciate the reviewer for kindly acknowledging the novelty of our method, in particular, the core idea of diffusion-induced coarse-to-fine token ordering for vanilla autoregressive modeling.
>
> **Q0. Misleading "SOTA" Classification.**
>
> Thank you for pointing out the confusion here. We double-check the text and ensure that the "leading" claim is only restricted within vanilla AR comparison, and we will revise the wording to make this scope explicit and the corresponding claims more rigorous and precise.
>
> **Q1. Can a direct inference latency benchmark (e.g., sec/image) be provided?**
>
> The K=8 diffusion steps only require a minor time to decode discrete codes into image pixels. We use stronger LlamaGen models [1] as vanilla AR model baselines. All models are benchmarked with batch size 8, BF16 precision, without compilation, over 10 forward passes on a single A100 GPU. KV cache is enabled in AR models. We include DiT-XL with 8 sampling steps (per request) and 250 steps (original) for comparison. The wall time latency comparison is as follows:
>
> | model                | #tokens/#steps | gFID  | generative model time   | tokenizer/VAE decoding time | total time | decoding % |
> | -------------------- | ------- | ---- | --------- | ----------------------- | ---------- | ---------- |
> | stronger LlamaGen-L  | 256     | 2.20 | 5.2372 s  | 0.0485 s                | 5.2858 s   | 0.92%      |
> | D-AR-L               | 256     | 2.44 | 5.3640 s  | 0.4470 s                | 5.8109 s   | 7.69%      |
> | stronger LlamaGen-XL | 256     | 2.16 | 7.8130 s  | 0.0510 s                | 7.8639 s   | 0.65%      |
> | D-AR-XL              | 256     | 2.09 | 7.8320 s  | 0.4478 s                | 8.2798 s   | 5.41%      |
> | DiT-XL               | 8       | ---  | 0.3277 s  | 0.0599 s                | 0.3876 s   | 15.5%      |
> | DiT-XL               | 250     | 2.27  | 9.8862 s | 0.0580 s                | 9.9441 s   | 0.58%      |
> | D-AR-XXL             | 256     | 2.00 | 10.4178 s | 0.4490 s                | 10.8668 s  | 4.13%      |
>
> For all D-AR variants, the diffusion-based decoder accounts for only 4–8% of the total wall-clock time, confirming that the sequential diffusion decoding overhead is modest compared to the AR generation process.
>
>
> **Q2. How is this coarse-to-fine tokenization paradigm expected to scale to high-resolution (1024x1024) T2I tasks?**
>
> Due to the limited high-resolution data and hardware availability during the rebuttal period, we only conduct experiments on 512x512 ImageNet training samples. Surprisingly, we find that we could finetune the sequential diffusion decoder solely to a higher 512x512 resolution with only 256 tokens to reach competitive performance, without touching the tokenizer encoder and the corresponding discrete token space in the D-AR tokenizer. We evaluate the 50K generated samples against 512x512 reference samples following the ADM evaluation pipeline. The 512x512 experiment results are as follows:
>
> | model   | FID$\downarrow$     | IS$\uparrow$ | Precision$\uparrow$ | Recall $\downarrow$|
> | -------- | -------- | -------- | ------------- | ------------- |
> | DiT-XL | 3.04   | 240.82       | 0.84        | 0.54        |
> | SiT-XL | 2.62   | 252.21       | 0.84          | 0.57        |
> | D-AR-L | 2.60   | 265.33       | 0.81          | 0.58        |
>
> The qualitative samples are also available at [this anonymous link](https://anonymous.4open.science/r/dar-samples-5E94/).
>
> These results indicate that token-efficient decoder tuning scales well to higher resolution: we keep the number of tokens fixed (256) and let the diffusion decoder absorb the additional spatial complexity. This strongly suggests the possibility that for 1024×1024 we only need compact or mildly increased token counts, and can continue to delegate fine-grained spatial detail to the diffusion process in the D-AR tokenizer, rather than scaling the token sequence length quadratically with image resolution.

---

> > ### Author Response · Authors · 2025-11-27
> >
> > **Q3. The ablation on K (number of groups) in Figure 7 seems counterintuitive.**
> >
> > The discrepancy between rFID and gFID across different K is, in fact, expected once we separate the reconstruction and generation sides.
> >
> > As we increase K (the number of groups in the sequential diffusion), two things happen:
> >
> > 1. *Reconstruction (rFID) becomes harder with the larger K.*
> >
> > - With more groups, the diffusion decoder can only condition on fewer tokens each step compared to the less group variant, given the fixed total token budget. Therefore, the worst rFID is expected with the larger K, especially the K=16 strongest linearization variant, which gives the worst rFID. In contrast, the K=1 no linearization variant, the diffusion decoder can condition on full 256 tokens each step and therefore is expected to yield the best reconstruction rFID.
> >
> > 1. *Generation becomes easier for the AR modeling at large K. **However, gFID is also bounded by rFID.***
> > - With more groups, the coarse-to-fine linearization becomes stronger for the token ordering and therefore eases the token autoregressive modeling, which is reflected in the AR training loss curves in Figure 7. **However, at the same time, gFID is fundamentally bounded by the tokenizer reconstruction rFID.** Therefore, the K=16 strongest linearization variant has worse gFID given the large rFID gap behind other K variants. This does not contradict our conclusion since one can find that despite variants with K from 1 to 8 yield progressively worse rFIDs, their gFIDs remain essentially at the same or even slightly better level.
> >
> >
> > **Q4. What is the distribution of "work" between the 1.4B AR model and the 300M Tokenizer?**
> >
> > a) **What if ground-truth tokens are provided for the first 50% of the sequence and the AR model only generates the last 50%?**
> >
> > This is a good insight and valuable suggestion here! We conduct experiments based on your suggested settings here: given the heading H tokens from the tokenizer, we use D-AR-L to generate the remaining tailing 256-H tokens based on the first H tokens as the context. We evaluate the 50K images in the ImageNet validation set and benchmark them against ground-truth images with rFID:
> >
> > | H   | rFID$\downarrow$     |
> > | -------- | -------- |
> > | 64, 25% | 1.99   |
> > | 128, 50% | 1.65   |
> > | 192, 75% | 1.55   |
> > | 256, tokenizer upper bound | 1.52   |
> >
> > As shown above, when provided with more leading tokens, the AR model can accurately predict the remaining tokens and preserve the reconstruction given the leading tokens as context. This suggests that the leading tokens carry more critical information than the trailing ones, which is consistent with our coarse-to-fine linearization token assumption.
> >
> > b) **What if the AR model only generates the first 50% of tokens (z_1...z_128)?**
> >
> > One way to support the first 128 tokens for D-AR without fine-tuning is to directly perform jump-estimation of image pixels once a sufficient number of tokens have been generated; the corresponding results are reported in Table 5 of the appendix. We also fine-tune only the sequential diffusion decoder to accept the first 128 tokens produced by the frozen tokenizer encoder, i.e., these 128 tokens are trained to represent the full diffusion trajectory in pixel space. The rFID of 128 token tokenizer is 1.79.
> >
> > | model                | #tokens | gFID |
> > | -------------------- | ------- | ---- |
> > | D-AR-L               | 256     | 2.44 |
> > | D-AR-L, jump-estimate| 128     | 3.94 |
> > | D-AR-L, 128 finetuned| 128     | 2.56 |
> >
> > We provide a qualitative comparison using the same generated token sequences (the 128-finetuned variant uses only the first 128 tokens) at [this anonymous link](https://anonymous.4open.science/r/dar-samples-5E94/). The D-AR model can still produce an acceptable image using the first 128 tokens despite local details varying mildly from the full token counterpart.
> >
> > **Summary**
> >
> > The experimental results show that the last 50% tokens focus more on local details than global semantics/critical information compared to the first 50% tokens. This, again, confirms our diffusion-induced token sequence to follow the coarse-to-fine ordering. We can conclude that the work of the trained AR models mainly lies in the image semantics/global layouts, and the trained tokenizer mainly focuses on the low-level detail modeling given a fixed token sequence. This conclusion can also be validated by the high-resolution experiment in response to Q.2.

---

> > > ### Author Response · Authors · 2025-11-27
> > >
> > > **Q5. Is the "streaming preview" advantage overstated compared to standard diffusion?**
> > >
> > > Thank you for pointing that out here. In the paper, we do **not** state that the preview ability of the D-AR framework is an advantage over the standard diffusion. Rather, our point is that the preview is an intriguing property of D-AR models **within the autoregressive modeling paradigm**, since most AR models, especially raster-scan ones, do not naturally support such preview ability, but D-AR can do so by design thanks to its diffusion-aligned token ordering. In other words, we are highlighting that D-AR brings a diffusion-like preview behavior into the AR setting, where this functionality is typically absent, **not highlighting** that it outperforms standard diffusion in this regard.
> > >
> > > **Q6. Does fine-tuning this way waste the pre-trained LLM's capabilities?**
> > >
> > > Thank you for the thoughtful question. We would like to clarify that we **only adopt standard Llama architecture** for autoregressive visual models and **train our D-AR models from scratch** with randomly initialized parameters, where we do not use language weights in experiments in the paper. The main research scope of this paper is to find better autoregressive modeling approaches for visual generation with the vanilla autoregressive transformer. Therefore, Stage 2 training does not destroy any existing language capabilities since we start from scratch. We will revise the related text to clearly state that we use **the architecture only**, not a pre-trained LLM.
> > >
> > > Conceptually, we agree with your point that if one were to start from a strong pretrained LLM and then fine-tune it only on visual tokens, then its language capabilities could be degraded. We agree that a very important and practical research direction is to reuse pretrained LLMs for visual generation while preserving their language capabilities. Existing works suggest that a promising approach is to train on a mixture of language-only tokens and text–image tokens with a carefully chosen mixing ratio, as explored in [1,2,3]. This can yield unified multimodal models that maintain most language capabilities.
> > >
> > > Our D-AR framework is compatible with this unified multimodal path: one could initialize from an LLM pretrained on a language corpus, then finetune it with mixed text + image tokens in future work, leveraging both the language/knowledge prior and our visual modeling improvements for vanilla AR models. Exploring this direction is beyond the scope of the current paper, and we leave it as a natural further work.
> > >
> > > [1] Visual Instruction Tuning. NeurIPS 2023.
> > >
> > > [2] Emu3: Next-Token Prediction is All You Need. arXiv 2024.
> > >
> > > [3] Emerging Properties in Unified Multimodal Pretraining. arXiv 2025.

---

> ### Comment · Reviewer_iSt8 · 2025-11-28
>
> Thanks the authors for the detailed explanation and insightful discussion. My concerns and questions have been well resolved. I remain positive about this paper.

---

### Official Review · Reviewer_q2wL · 2025-10-31

**Soundness:** 3
**Presentation:** 3
**Contribution:** 2
**Rating:** 4
**Confidence:** 4

**Summary:**

This paper introduces D-AR, a framework bridging pixel-level diffusion and autoregressive modeling for image generation. The core idea is to transform 2D images into 1D discrete token sequences using a sequential diffusion tokenizer, where tokens are ordered in a coarse-to-fine manner corresponding to diffusion steps. A standard decoder-only Transformer is then used to perform autoregressive next-token prediction on these tokens. Sequential token generation directly mirrors diffusion denoising steps on pixels, enabling KV cache-friendly inference, streaming pixel decoding with consistent previews, and zero-shot layout-controlled synthesis.

**Strengths:**

1. Novel framework: Integrates diffusion and autoregressive modeling, providing a unified approach that preserves the strengths of both paradigms.
2. Sequential diffusion tokenizer: Generates coarse-to-fine token sequences corresponding to diffusion steps, naturally suited for AR modeling.
3. Preserves vanilla AR architecture: Works with standard decoder-only Transformers without modifications to causal masks, attention, or training schemes. Supports streaming pixel decoding, consistent previews, and zero-shot layout-controlled synthesis.

**Weaknesses:**

1. Dataset and task limitation: Experiments are only on ImageNet 256×256 class-conditional generation. High-resolution images, other datasets (COCO, LSUN, FFHQ), or tasks (image repair, image edit) are not tested.
2. Model complexity and inference speed: The sequential diffusion tokenizer adds 300M parameters, and AR generation requires predicting 256 tokens sequentially. Although KV caching and token grouping mitigate some overhead, more tokens still increase generation latency compared to smaller token setups like Titok-S (128 tokens), which may limit real-time applications. It would be interesting to know whether the authors have tried using Titok-S or a smaller token budget within the D-AR framework.

**Questions:**

Have you compared the differences in reasoning speed or generation throughput between D-AR and other  AR methods ?

---

> ### Author Response · Authors · 2025-11-27
>
> Thank you for acknowledging the novelty and design philosophy of our work. We sincerely value your comments here. We address your concerns as follows:
>
> **Q1. Dataset and task limitation.**
>
> We conduct experiments here with high-resolution image generation according to your suggestion. Our main idea here is to fix the parameters of the autoregressive model (D-AR-L w/ 256 tokens) and only finetune the sequential diffusion decoder of the D-AR tokenizer to a higher resolution. We evaluate the 50K generated samples against 512x512 reference samples following the ADM evaluation pipeline. The 512x512 experiment results are as follows:
>
> | model   | FID$\downarrow$     | IS$\uparrow$ | Precision$\uparrow$ | Recall $\downarrow$|
> | -------- | -------- | -------- | ------------- | ------------- |
> | DiT-XL | 3.04   | 240.82       | 0.84        | 0.54        |
> | SiT-XL | 2.62   | 252.21       | 0.84          | 0.57        |
> | D-AR-L | 2.60   | 265.33       | 0.81          | 0.58        |
>
> This competitive result also shows the flexibility of the D-AR framework with only 256 tokens since we only fine-tune the sequential diffusion decoder in D-AR to the higher resolution. We also visualize qualitative samples at [this anonymous link](https://anonymous.4open.science/r/dar-samples-5E94/)
>
> **Q2. Model complexity and inference speed.**
>
> **Speed comparison**
>
> Thank you for your valuable suggestions. We here compare the speed of AR models. We use stronger LlamaGen models [1] as vanilla AR model baselines. All models are benchmarked with batch size 8, BF16 precision, without compilation, over 10 forward passes on a single A100 GPU. KV cache is enabled in AR models.
>
> | model                | #tokens | gFID  | AR time   | tokenizer decoding time | total time | decoding % |
> | -------------------- | ------- | ---- | --------- | ----------------------- | ---------- | ---------- |
> | stronger LlamaGen-L  | 256     | 2.20 | 5.2372 s  | 0.0485 s                | 5.2858 s   | 0.92%      |
> | D-AR-L               | 256     | 2.44 | 5.3640 s  | 0.4470 s                | 5.8109 s   | 7.69%      |
> | stronger LlamaGen-XL | 256     | 2.16 | 7.8130 s  | 0.0510 s                | 7.8639 s   | 0.65%      |
> | D-AR-XL              | 256     | 2.09 | 7.8320 s  | 0.4478 s                | 8.2798 s   | 5.41%      |
> | D-AR-XXL             | 256     | 2.00 | 10.4178 s | 0.4490 s                | 10.8668 s  | 4.13%      |
>
> As shown above, the sequential diffusion decoder in the D-AR framework accounts for **less than 8%** of the total wall-clock time during image generation and therefore does not introduce unacceptable additional latency. Moreover, as the D-AR model size increases, the amortized latency cost of decoding into images becomes smaller.
>
>
> **Smaller token budget variant**
>
> Thank you for your valuable advice. One way to support smaller token budgets for D-AR without fine-tuning is to directly perform jump-estimation of image pixels once a sufficient number of tokens have been generated; corresponding results are reported in Table 5 of the appendix. Building on this idea, we instead fine-tune only the sequential diffusion decoder to accept the first 128 tokens produced by the frozen tokenizer encoder, i.e., these 128 tokens are trained to represent the full diffusion trajectory in pixel space. In this way, we do not need to retrain or finetune AR models here. We term this as _128 finetuned_ D-AR. The rFID of 128 finetuned D-AR tokenizer is 1.79, against 1.52 rFID of the D-AR tokenizer with full 256 tokens. The gFID and inference time results are:
>
> | model                | #tokens | gFID  | AR time   | tokenizer decoding time | total time |
> | -------------------- | ------- | ---- | --------- | ----------------------- | ---------- |
> | D-AR-L               | 256     | 2.44 | 5.3640 s  | 0.4470 s                | 5.8109 s   |
> | D-AR-L, jump-estimate| 128     | 3.94 | 2.7122 s  | 0.2252 s                | 2.9374 s   |
> | D-AR-L, 128 finetuned| 128     | 2.56 | 2.7306 s  | 0.4619 s                | 3.1926 s  |
>
> We see that the 128-finetuned D-AR-L variant achieves a competitive gFID while using only about **55%** of the original wall-clock time. We also provide a qualitative comparison using the same generated token sequences (the 128-finetuned variant uses only the first 128 tokens) at [this anonymous link](https://anonymous.4open.science/r/dar-samples-5E94/).
> These findings show the flexibility of the D-AR framework and further suggest that there is potential to finetune D-AR into even faster variants with smaller budgets.
>
> [1] RandAR: Decoder-only Autoregressive Visual Generation in Random Orders. CVPR 2025.

---

### Official Review · Reviewer_MQZQ · 2025-10-31

**Soundness:** 2
**Presentation:** 3
**Contribution:** 2
**Rating:** 4
**Confidence:** 4

**Summary:**

The authors propose D-AR, which reformulates the pixel diffusion process as a vanilla autoregressive procedure. They introduce a diffusion tokenizer that organizes discrete image tokens in a coarse-to-fine order. A conventional next-token prediction framework is then applied to sequentially generate these tokens, which are subsequently grouped and fed into the diffusion detokenizer as conditions to decode images.

**Strengths:**

1. The idea to project diffusion process into a vanilla autoregressive procedure is interesting.
2. The paper is well-written and easy to follow.

**Weaknesses:**

1. Although the idea is interesting, I am not entirely convinced that this approach truly demonstrates a synergistic effect between autoregressive modeling and diffusion. The autoregressive component still relies on discrete tokens, which are then used as conditions for the diffusion model. Consequently, the information loss inherent in the discrete tokens is propagated into the diffusion process. This is evidenced by the higher rFID of the diffusion tokenizer compared to its continuous VAE counterpart.
2. The results in Table 3 are not a fair comparison with other methods in terms of parameter count. The diffusion tokenizer itself contains an additional 300M parameters, whereas the tokenizers in other approaches (whether continuous or discrete) are much smaller. For instance, D-AR-XL actually has around 1.1B parameters, yet achieves an FID of 2.09, which is worse than the approximately 700M SiT-XL diffusion model with an FID of 2.06. This raises concerns about the advantage of D-AR compared with a pure diffusion model under the same parameter budget.
3. The authors claim that an intriguing property of their model is the ability to preview the diffusion generation process. However, this is not unique to D-AR, but rather a general characteristic of diffusion models in general.

**Questions:**

Please refer to the Weakness section.

---

> ### Author Response · Authors · 2025-11-27
>
> We would like to thank the reviewer for acknowledging the presented idea and the writing of our work. We here, however, feel it necessary to clarify the context that our initial motivation is to make vanilla autoregressive models to have better generative modeling for images, where several challenges are as follows:
>
> 1) Discrete tokens impose a very compact space for latent tokens, which are limited in the number of values (that is, discrete), especially when compared to continuous latents by continuous VAEs.
>
> 2) How the token ordering by discrete tokenizers on images (naturally 2D) should be defined for vanilla autoregressive modeling (naturally 1D), as discussed in line 073-077 and 157-161 in the paper.
>
> We address your specific concerns as follows:
>
> **Q1. Synergistic effect between autoregressive modeling and diffusion. The loss is propagated into the diffusion process.**
>
> First, we would like to address again that the quantization error arising from discrete visual codes is inherent to general vector quantization operations in discrete visual tokenizers (see Challenge 1 above) and is not unique to our methods. Our main goal is to improve the generative quality of autoregressive models, rather than focusing on reconstruction performance only, which could be inferred from slightly worse reconstruction rFID but on-par generation gFID compared to SD-VAE and DiT/SiT. These rFID and gFID contrasts also suggest, on the contrary, that the information loss in the discrete tokens could be remedied by the diffusion process in the D-AR tokenizer in the image generation process. Again, the synergistic effect of the proposed D-AR between autoregressive and diffusion lies in the diffusion-induced token ordering, which is better for autoregressive modeling and leads to our competitive gFID scores among vanilla autoregressive peers.
>
> **Q2. Concerns about the tokenizer parameter count.**
>
> We fully understand your concern, as we follow the convention to compare parameter counts only in generative models in the Table. We will clarify this further in the future version. Regarding the parameter count itself, we address it on two sides:
>
> a) __Why large parameters are needed in the D-AR tokenizer?__
>
> The reason behind large parameters is that we use a diffusion decoder to decode latent codes into image pixels, which is different from other tokenizers/VAEs that use CNN decoders. The diffusion decoder, per se, especially pixel diffusion transformers, needs large parameter counts to well model the diffusion process on raw image pixels [1, 2] (to be more specific, see Figure 2 in [1] and Table 6 in [2]). Actually, the parameter count of the D-AR tokenizer is modestly set to be small, compared to tokenizers based on pixel diffusion decoders (e.g.,epsilon-VAE has up to 650M+ parameters, and FlowMo has 517M and 947M parameters [2])
>
> [1] Epsilon-VAE: Denoising as Visual Decoding, ICML 2025
>
> [2] Flow to the Mode: Mode-Seeking Diffusion Autoencoders for State-of-the-Art Image Tokenization, ICCV 2025
>
> b) __What comes if we reduce the parameter count?__
>
> We also reduce the parameter of the D-AR diffusion decoder from 185M to 62M (changed by the width and depth of the pixel diffusion transformer to 512 and 8, respectively) and maintain the D-AR tokenizer encoder and the code space unchanged.
> We finetune the sequential diffusion decoder till convergence and name it `light tokenizer` in the following table.
>
> | model | rFID | gFID |
> | ----- | ----- | ----- |
> | original tokenizer + D-AR-L | 1.52 | 2.44 |
> | light tokenizer + D-AR-L | 2.65 | 2.75 |
>
> We can find that the large diffusion decoder parameter count is crucial to the reconstruction performance but less vital to the generation score. This suggests strongly that we could design and distill lightweight sequential diffusion decoders once we have a diffusion-induced discrete token space.
>
> **Q3. The preview is a general characteristic of diffusion models.**
>
> In the paper, we state that the D-AR framework can leverage characteristics of both diffusion models and autoregressive models, where the streaming preview, of course, is the ability inherited from the diffusion world.
> We state that the preview is an intriguing property of D-AR models **in the context of autoregressive modeling**, since most AR models, especially raster-scan ones, do not support this preview ability, and D-AR can perform this by design thanks to its diffusion-aligned token ordering.
> Also, the preview ability of D-AR can be performed at no cost since it directly operates on pixels and does not require an extra VAE/tokenizer decoder forward.
> We will refine the relevant sentences to further clarify this point.

---

### Official Review · Reviewer_XBux · 2025-11-03

**Soundness:** 3
**Presentation:** 3
**Contribution:** 3
**Rating:** 6
**Confidence:** 2

**Summary:**

The paper introduces D-AR (Diffusion via Autoregressive models), a new framework that reinterprets the image diffusion process as a standard autoregressive next-token prediction task by using a specially designed sequential diffusion tokenizer that maps coarse-to-fine diffusion steps into a linear sequence of discrete tokens. This approach enables high-quality image generation with vanilla autoregressive transformers (e.g., Llama backbones), achieving state-of-the-art FID scores (2.00 with 1.4B parameters) on ImageNet while supporting streaming previews and zero-shot layout control—without modifying core AR mechanisms.

**Strengths:**

(1) Its sequential diffusion tokenizer naturally imposes a coarse-to-fine token ordering aligned with diffusion denoising steps, which is highly suitable for autoregressive modeling.
(2)  The framework supports streaming, consistent previews during generation at no extra cost by leveraging diffusion’s ability to jump-estimate final images from partial token sequences.
(3) It enables zero-shot layout-controlled synthesis by fixing prefix tokens, all without finetuning.
(4) D-AR benefits from the efficiency of AR inference (e.g., KV caching) and operates directly on raw pixels without VAEs, simplifying the pipeline while maintaining compatibility with existing LLM infrastructure.

**Weaknesses:**

(1) I find D-AR to be a very interesting framework that combines two well-established paradigms. However, the motivation for introducing this new hybrid paradigm is unclear—current diffusion-based or purely autoregressive (AR) approaches already perform very well in image generation. Why is this fused paradigm necessary? What specific problem does it solve?
(2) D-AR cannot be easily integrated with state-of-the-art step distillation methods, such as one-step image generation. If step distillation is applied to reduce the number of sampling steps to just 1 or 2, what is the purpose or benefit of the autoregressive component in D-AR?

**Questions:**

see above

---

> ### Author Response · Authors · 2025-11-27
>
> Thank you for acknowledging the strengths of our work. We sincerely value your review and insights here. We address your concerns as follows:
>
> **Q1. Is the necessity of the proposed fused paradigm? What problem does it solve?**
>
> We would like to first address that the performance of current (purely or vanilla) autoregressive models for vision generative modeling still lags behind diffusion models in image quality. This motivates researchers to find better visual tokenizers that provide better discrete representations/orders for autoregressive modeling and minimize the gap between autoregressive models and diffusion counterparts. This is different from continuous VAEs: discrete tokenizers must operate in a very compact representation space (discrete tokens).
>
> In this context, our goal is to design a visual tokenizer that (1) achieves strong reconstruction quality and (2) induces a token ordering that is well-suited for vanilla autoregressive modeling.
> Diffusion models provide a natural starting point for this, which leads to our proposed D-AR tokenizer:
>
> Specifically,
> 1) For a fixed latent space (the same number of tokens and codebook size), our proposed sequential diffusion decoder outperforms LlamaGen VQGAN counterparts, where the latter is with a conventional CNN decoder, as shown in Table 1 in the paper. More specifically, the tokenizer can delegate reconstructing details to the diffusion decoder and learn to better tokenize an image into a compact discrete space.
>
> 2) Diffusion naturally provides a coarse-to-fine ordering, which benefits AR modeling, as shown in Figure 4 and Table 7 in the paper. This helps the AR transformer model better learn visual generation while maintaining the vanilla next token prediction.
>
> Overall, we aim to design an AR model to achieve “diffusion quality”, but the underlying mechanism is autoregressive with the next token prediction with discrete visual tokens.
>
> **Q2. The purpose or benefit of the autoregressive component in D-AR when the diffusion step is distilled into 1 or 2.**
>
> We truly appreciate this question on step distillation and one-step image generation. It is worth noting that our current diffusion steps are configured by default to 8, which is only a few in the context of diffusion sampling. That said, our method is, by design, orthogonal to the distillation and also compatible with current distillation methods. One distillation way is to finetune the sequential diffusion decoder with each step, with 256/N tokens as inputs, where N is the number of distilled steps. The autoregressive component in D-AR continues to play a central role in visual sequences when we distill by this method, since the mechanism of AR modeling coarse-to-fine token structure by diffusion is untouched and still benefits from our tokenizer pretraining with timestep corresponding grouped tokens.

---

### Meta-Review · Area_Chair_eDX4 · 2026-01-07

**Summary:**

The paper introduces D-AR, a novel framework that combines diffusion and autoregressive (AR) modeling for image generation. By reformulating the multi-step diffusion process as a single autoregressive next-token prediction task, the paper proposes a unique sequential diffusion tokenizer that enhances the image generation process.

Reviewers expressed varied opinions on the paper's contribution, methodology, and results. While some reviewers praised the innovative unification of diffusion and AR models, others raised concerns regarding the true novelty of the method, the clarity of the results, and the scalability of the framework. Despite the differing perspectives, the authors addressed most of the concerns raised and provided sufficient clarifications.

The paper is considered strong in its category, though improvements in terminology and experimental evaluation scope are recommended. Thus, the paper is recommended for acceptance.

**Reviewer Concerns:**

- Reviewer XBux raised concerns about the motivation behind combining AR and diffusion models and compatibility with step distillation. The authors clarified their approach and addressed the concern.

- Reviewer MQZQ doubted the synergistic effect of AR and diffusion, particularly due to potential information loss from discrete tokens. The authors clarified that the focus is on generative quality, and showed how diffusion mitigates this issue.

- Reviewer q2wL noted the limited experimental scope and scalability concerns for high-resolution tasks. The authors provided additional experiments on 512x512 images and clarified scalability, though the limited scope remains a concern.

- Reviewer iSt8 pointed out the limited SOTA comparison and tokenizer complexity. The authors clarified terminology and addressed latency concerns with detailed benchmarks, showing the overhead is modest.

**Reviewer Scores:**

The overall scores are somewhat divided, with two reviewers suggesting ratings of 6 and two raising concerns that slightly lower the overall evaluation. While there are disagreements about the paper's novelty, experimental scope, and clarity, the authors have substantially addressed the majority of concerns raised. In light of these revisions and clarifications, the paper should be accepted.

---

### Decision · Program_Chairs · 2026-01-26

Accept (Poster)